# Pharmacological and Chemical Potential of *Spiranthes sinensis* (Orchidaceae): A Narrative Review

**DOI:** 10.3390/plants11131692

**Published:** 2022-06-27

**Authors:** Yu-Jen Kuo, Jin-Kuo Pei, Wen-Wan Chao

**Affiliations:** 1Department of Health Wellness and Marketing, Kainan University, 1 Kainan Rd., Shinshing, Luchu, Taoyuan 33857, Taiwan; kuoyj@mail.knu.edu.tw (Y.-J.K.); kimcopei@mail.knu.edu.tw (J.-K.P.); 2Department of Nutrition and Health Sciences, Kainan University, 1 Kainan Rd., Shinshing, Luchu, Taoyuan 33857, Taiwan

**Keywords:** *Spiranthes sinensis*, Orchidaceae, dihydro/phenanthrenes, 3,4-dihydroxybenzaldehyde, ferulic acid, flavonoid

## Abstract

Orchidaceae is one of the largest families of flowering plants with more than 27,000 accepted species, and more than 31,000–35,000 species are estimated to exist in total. The orchid *Spiranthes sinensis* (Pers.) Ames, having ornamental and medicinal value, is widely distributed throughout Asia and Oceania. *S. sinensis* (Shou Tsao) is also known as Panlongshen among the common folk herbs. It has a fleshy root similar to ginseng, and the entire plant is widely used in traditional Chinese medicine. Owing to overexploitation and habitat destruction in recent years, the wild population has become scarce. The traits of this species show obvious differences in different countries. In the Taiwanese climate, it flowers during the Ching Ming Festival, also called the ching ming tsao. Previous investigations into *S. sinensis* have revealed the presence of flavonoids, homocyclotirucallane, dihydrophenanthrenes, ferulic acid, and 3,4-dihydroxybenzaldehyde. Phenolic constituents of structural and biological interest, including phenanthrenes and flavonoids, have been isolated and identified from *S. sinensis*. This natural product possesses extensive bioactivity, including anti-tumor, anti-inflammatory, and antioxidant effects. In this review, we outline the herbal medicine formulations and plant-derived natural products of *S. sinensis*.

## 1. Introduction

Phenetic analysis of Orchidaceae was conducted by Clifford and Lavarack (1974). Orchidaceae is one of the largest families of flowering plants with more than 27,000 accepted species, and more than 31,000–35,000 species are estimated to exist in total. The Orchidoideae subfamily contains orchids with a single, erect, basitonic, fertile anther (monandrous). It is well known as a family in which wide crosses are possible; interspecific and intergeneric hybrids are the basis for a thriving commercial market [1]. Orchidaceae is also abundant in phenanthrene- and dihydrophenanthrene-type compounds [2,3].

The *Spiranthes* genus (Orchidaceae: Cranichideae, Orchidoideae) contains approximately 50 species that predominantly occur in North America and are widely distributed in Central and South America, Eurasia, and Australia. *Spiranthes* (∼36 species, Orchidaceae) is a small genus with a global distribution. It has a center of diversity in North America, with only a few species occurring in Asia [4]. *Spiranthes sinensis* (Pers.) Ames is a species of Orchidaceae, also known as Panlongshen among the common folk herbs, which has ornamental and medicinal value. In recent years, because of damage to the natural environment due to overdevelopment, vandalism, the wild collection of plants, and other factors associated with human activities, many precious plant resources, including this species, have become endangered. *S. sinensis* is widely distributed throughout Asia and Oceania [5]. The traits of the species show obvious differences in different countries. Owing to overexploitation and habitat destruction in recent years, it has become scarce in the wild. The entire plant of *S. sinensis*, its local name being ‘Shou Tsao’, is used in traditional Chinese herbal medicine [6,7]. *S. sinensis*, commonly known as qingming grass, is a terrestrial orchid endemic to Taiwan. The flowers are arranged in a spiral along a flowering stalk, and a pink color symbolizes joy and luck. The species is used as an ornamental plant and has the potential to be further developed for commercial horticultural purposes. 

Whole plants of *Spiranthes sinensis* (Orchidaceae) were collected from Kainan University, Taoyuan City, (121°15′00″ E, 25°03′00″ N), Taiwan, by Professor Yu-Jen Kuo. The plant material was harvested from April to May 2019 (Figure 1 and Figure 2). *S. sinensis* is a well-known Chinese herb and is an important medicinal plant. It is widely used in the treatment of inflammation, cancer, diabetes, and other diseases [7,8,9]. Traditional medicines derived from the bioactive compounds of plant rich sources play a key role in traditional healthcare and as a food source. Here, we present a narrative review of orchid-derived bioactive compounds, which will be helpful for novel drug discovery.

## 2. Bioactive Compounds from *Spiranthes sinensis* (Orchidaceae)

*Spiranthes sinensis* (Shou Tsao) is an orchid plant, and its fleshy root, similar to ginseng, is called Panlongshen. In the Taiwanese climate, it flowers during the Ching Ming Festival, also called the ching ming tsao. Studies have indicated that *S. sinensis* possesses anti-cancer, anti-inflammatory, and anti-bacterial effects. Recently, some experts have analyzed its chemical ingredients and have investigated the pharmacological activities of *S. sinensis*, and it has been reported that it contains phenanthrenes, flavones, and ferulic acid ester groups. These chemical ingredients exhibited excellent pharmacological activity. *Spiranthes sinensis* (Pers.) Ames is an orchid that is widely grown in China, Japan, and Taiwan and is often used in folk medicine [10,11].

Oxidative stress is an imbalance between the production of reactive oxygen species (ROS) and reactive nitrogen species (RNS) and their clearance by antioxidants. The prolonged generation of ROS caused by inflammatory mediators (pro-inflammatory cytokines, TNF-α, IL-1 β, and IL-6) can induce oxidative DNA damage [12]. Malondialdehyde (MDA), an indicator of lipid peroxidation, is frequently measured to determine the level of oxidative damage in cells [13]. Inducible nitric oxide synthase (iNOS) is usually associated with nitric oxide (NO) overproduction and can be upregulated in response to various pro-inflammatory cytokines (TNF-α, IL-1 β, and IL-6) in different cell types, such as macrophage cells. The regulation of inflammation-related enzymes, such as heme oxygenase-1 (HO-1), iNOS, and COX-2, plays an important role in inflammation [14].

An ethyl acetate fraction of *S. sinensis* extract (SSE) was used to reveal the anti-inflammatory properties of SSE, showing its involvement in the suppression of iNOS and pro-inflammatory cytokines by inducing the HO-1 pathway in LPS-stimulated murine macrophage cells (RAW264.7 cell line) and BALB/c mice [9].

### 2.1. Homocyclotirucallane and Dihydro/Phenanthrenes

*S. sinensis* is widely used in the treatment of inflammatory diseases, cancer, and chest disorders [7]. Previous investigations into *S. sinensis* have shown that they are a rich source of flavonoids, homocyclotirucallane, dihydrophenanthrenes, ferulic acid (FA), and 3,4-dihydroxybenzaldehyde (DHB) [5].

#### 2.1.1. Anti-Hepatoma Cell

*S. sinensis* (Orchidaceae) has been used as a folk medicine in Taiwan. A novel homocyclotirucallane, sinetirucallol, and two new dihydrophenanthrenes, sinensols G and H, were isolated from the aerial parts of *S. sinensis* [6]. An ethanolic extract of the aerial parts of *S. sinensis* was successively partitioned between n-BuOH and ethyl acetate. The ethyl acetate (EtOAc)-soluble fraction showed moderate (42.5%) anti-hepatitis B virus e-antigen (HBeAg) activity against the hepatoma cell line MS-G2 at 100 µg/mL. Column chromatography of the EtOAc fraction led to the isolation of five known and six novel dihydrophenanthrenes [10]. The traditional Chinese medicine ‘Panlongshen’ is derived from the dried root or the whole plant of *S. sinensis*, which is an Orchidaceae plant [15].

#### 2.1.2. Inhibits Adipogenesis in 3T3-L1 Cells

Structural and biological constituents, including phenanthrenes and flavonoids, were isolated and identified from *S. sinensis*. These types of natural plant compounds (NPCs) possess biological activities, including anti-tumor, anti-inflammatory, and antioxidant activities [16,17,18]. *S. sinensis* contains various phytochemicals such as phenanthrenes, flavonoids, coumarins, and steroids, of which phenanthrene is the major compound. Six phenanthrene derivatives, in the ethyl acetate fraction of *S. sinensis*, showed that sinensol-C could regulate adipogenesis via the downregulation of adipogenic transcription factors (peroxisome proliferator-activated receptor, PPAR) and the upregulation of AMP-activated protein kinase (AMPK) in 3T3-L1 adipocytes [19].

#### 2.1.3. Anti-Cancer Activity

Epithelial-to-mesenchymal transition (EMT) has been identified in human tissues in various cases of renal, pulmonary, and liver fibrosis [20]. EMT, as a source of myofibroblasts, is a reversible process in which epithelial cells lose their characteristics, such as E-cadherin, and gain invasive phenotypes, including N-cadherin, vimentin, and fibronectin [21]. It is a dynamic and reversible molecular mechanism with fundamental physiological roles in organ development and wound healing [22].

Cisplatin is a chemotherapeutic drug for treating melanoma, which also causes adverse side effects in cancer patients. Spiranthesphenanthrene, isolated from *Spiranthes sinensis*, is cytotoxic towards SGC-7901 gastric cancer cells, HepG2 hepatocellular carcinoma, and B16-F10 melanoma tumor cell lines. Spiranthesphenanthrene A exhibited higher cytotoxic activity toward the murine B16-F10 melanoma cell line than the chemotherapy drug cisplatin (IC_50_ = 19.0 ± 7.3 μM). It also inhibited B16-F10 cancer cell migration, which might be related to its ability to inhibit EMT. Thus, phenanthrenes may be the active natural plant compounds responsible for the anti-cancer activity of *S. sinensis* [18]. A large number of differently substituted phenanthrenes with biological activity have been isolated from the Orchidaceae family. Natural phenanthrenes can occur in a monomeric form and in many cases present only hydroxy and/or methoxy-substitutions. Phenanthrene and its derivatives are currently the most diverse and characteristic compounds found in medicinal orchid plants. Bioactive compounds from medicinal plants or from herbal diets, namely dietary phytochemicals, have lately been shown to play a significant role in cancer chemoprevention and are leading to the development of a new and alternative method of cancer prevention and therapy.

### 2.2. 3,4-Dihydroxybenzaldehyde (3,4-DHB)

*S. sinensis* is an herb used by Taiwanese indigenous people as a folk medicine. It is used to cook rice or boil ginseng dragon chicken for the treatment of some diseases. *S. sinensis*, commonly known as the Chinese Spiranthes, is a species of orchid that occurs in eastern Asia, west of the Himalayas, in southern and eastern New Zealand, and in northern Siberia. Previous phytochemical investigations of *S. sinensis* have shown the presence of 3,4-dihydroxybenzaldehyde (3,4-DHB). 3,4-DHB, also known as protocatechualdehyde, is a water-soluble polyphenolic antioxidant. Additionally, 3,4-DHB possesses anti-inflammatory activities, neuroprotective effects against oxidative stress, and anti-cancer activities [23]. 3,4-DHB is a structurally simple polyphenolic compound found in grapevine leaves, buckwheat, barley, green cavendish bananas, and Chinese red sage plants [24,25,26]. It is widely used as a food flavor and pharmaceutical intermediate. It possesses neuroprotective effects in rat models of middle cerebral artery occlusion/reperfusion and lipopolysaccharide (LPS)-treated BV2 microglial cells [27]. It also exhibits anti-microbial properties and is commonly used to treat diarrhea. It minimizes hydrogen peroxide (H_2_O_2_)-, superoxide-, and peroxyl-radical-mediated oxidative damage and apoptosis. The plant antioxidant 3,4-dihydroxybenzaldehyde (3,4-DHB) protected against pentachlorophenol (PCP)-induced cytotoxicity and genotoxicity in human red blood cells (RBC) and lymphocytes [28]. Anjum et al. demonstrated that 3,4-dihydroxybenzaldehyde (3,4-DHB) significantly and dose-dependently mitigated sodium fluoride (NaF)-induced oxidative modifications [29].

### 2.3. Ferulic Acid (FA)

#### 2.3.1. Anti-Inflammatory Activity

Ferulic acid (4-hydroxy-3-methoxycinnamic acid) was first isolated from a commercial resin in 1866 before being chemically synthesized in 1925 as a phenolic acid. It is commonly present in plants and is a bioactive compound in many foods, including cereals, vegetables, fruits, and drinks. It is also used as a food additive in many countries. It acts as a potent antioxidant by scavenging free radicals via its phenolic nucleus and extended side-chain conjugation and forms a resonance-stabilized phenoxyradical [30]. FA also inhibits LPS-induced reactive oxygen species (ROS) production, nitrite accumulation, and iNOS protein expression [31,32,33].

#### 2.3.2. Apoptosis Pathway

Reactive oxygen species (ROS) activate the intrinsic or mitochondrial apoptosis pathway. Excessive ROS production may lead to the loss of the mitochondrial membrane potential (MMP, ΔΨm) and the activation of the mitochondrial apoptotic pathway. A reduction in MMP may activate the intrinsic apoptotic pathway by affecting factors such as cytochrome c, Apaf-1, and caspase-9, thereby changing the permeability of the mitochondrial membrane and activating the B-cell lymphoma 2 (Bcl-2) and caspase families [34]. Many anti-cancer agents have been reported to mediate their activity through mitochondrial events such as the depletion of glutathione, the production of reactive oxygen species, and the activation of intrinsic apoptosis pathways (for example, cytochrome c, Apaf-1, and caspase-9).

FA (4-hydroxy-3-methoxycinnamic acid, C_10_H_10_O_4_) is found in various fruits, vegetables, natural herb extracts, spices, and coffee. The plant-derived polyphenolic compound FA has also demonstrated the efficacy of cinnamic acid derivatives (such as ferulic acid, caffeic acid, and chlorogenic acid) in the treatment of cancer [35]. FA induces osteosarcoma cell apoptosis through a high Bax/Bcl-2 ratio, causing the release of cytochrome c from mitochondria into the cytosol and leading to the activation of caspases-9 and -3, which then induce apoptosis. Similarly, FA can induce an increase in the expression of Bax, caspase-3, and caspase-9 in MIA PaCa-2 human pancreatic cancer cells [36]. Luo et al. demonstrated that FA has cytotoxic effects on HeLa and CaSki cervical carcinoma cells [37].

#### 2.3.3. Neuroprotective

FA, a hydroxycinnamic acid, is an abundant phenolic phytochemical. Lin et al. reported that FA reduced hypoxic injury in PC12 cells by preventing p38 MAPK, caspase-3, and COX-2 activation [38]. Gu et al. stated that 10 μM FA demonstrated both neuroproliferative and neurodifferentiative effects [39]. Nakayama et al. demonstrated that FA inhibits H_2_O_2_-induced apoptosis by regulating intracellular ROS levels and mitochondria-dependent pathways in PC12 cells [40]. Accumulated evidence has revealed that neurodegenerative diseases are due to ROS accumulation, which leads to lipid peroxidation, DNA damage, and neuronal injury or death.

#### 2.3.4. Anti-Nephrotoxicity and Anti-Diabetes

FA, a phenolic substance widely present in plants, is an important active component in many traditional Chinese medicines. FA ameliorated cisplatin-induced nephrotoxicity in rats [41]. FA can block the activation of smad2/smad3 signaling, upregulate the expression of E-cadherin, inhibit TGF-β1, and drive the EMT process in renal tubular epithelial cells (NRK-52E) [42]. FA and astragaloside IV synergistically reduced the expression of fibronectin, α-smooth muscle actin (α-SMA), and TGF-β1 and increased the production of NO in rats who underwent unilateral ureteral obstruction (UUO) [43]. FA had a protective effect against bone loss in ovariectomized rats [44]. Asano et al. [45] reported the therapeutic potential of FA against chronic cerebral hypoperfusion-induced swallowing dysfunction in animals. Balasubashini et al. [46] also demonstrated that FA prevents lipid peroxidation in rats with streptozotocin (STZ)-induced diabetes.

#### 2.3.5. Anti-Hepatic Fibrosis

The hepatotoxicity of carbon tetrachloride (CCl_4_) is mediated by the highly reactive trichloromethyl free radical (CCl_3_•) and/or the peroxyl radical (CCl_3_OO•), activated forms of CCl_4_ formed by the action of the cytochrome P450 system, including CYP2B1, CYP2B2, CYP2E1, and possibly CYP3A. The formation of these free radicals results in lipid peroxidation, covalent binding to macromolecules, and ultimately, cell necrosis [47,48,49]. Experimental animals with CCl_4_-induced liver injury are commonly used for screening hepatoprotective/anti-hepatotoxic drugs [50,51,52]. FA improved TGF-β1-induced hepatic fibrosis via the regulation of the TGF-β1/Smad pathway. All CCl_4_-induced changes were markedly attenuated by FA treatment [53].

#### 2.3.6. Synergistic Effects

Combination therapy involving different therapeutic strategies mostly provides rapid and effective results as compared to monotherapy in diverse areas of clinical practice. Herbal medicines and plant-derived natural products have historically been a major source of anti-cancer drugs. The herbal combination chemotherapies also showed a significant reduction in the undesirable side effects of synthetic-drug-based chemotherapies [54]. FA is a phenolic compound with potent antioxidant activity. FA has potent antioxidant, cardioprotective, and anti-inflammatory activities; it may be administered along with the anti-tumor agent doxorubicin as an adjunct to reduce its toxicity [55]. FA inhibits cisplatin-induced cytotoxicity by preventing ROS formation and inducing the production of endogenous antioxidants, indicating that it might be used as a protective agent against cisplatin-induced ototoxicity in House Ear Institute-Organ of Corti 1 (HEI-OC1) cells [56]. It is a bioactive substance with multiple functions such as having an antioxidant effect and being anti-inflammatory. It has a protective effect against renal and cardiovascular diseases [57,58]. Mahmoud et al. demonstrated that FA can upregulate the signal transduction of PPAR γ and Nrf2 in methotrexate (MTX)-induced nephrotoxicity and prevent the excessive production of ROS. FA can also inhibit the NF-κ B/NLRP3 inflammasome axis and apoptosis of renal cells by activating PPAR γ [59].

### 2.4. Flavonoids

Flavonoids are naturally occurring polyphenols that are ubiquitous in fruits, vegetables, and teas as well as in most medicinal plants. Flavonoids and phenolics are considered potent antioxidants owing to the presence of hydroxyl groups and conjugated ring structures, which scavenge free radicals via hydrogenation or complexation. Flavonoids have the basic skeletal structure of C_6_-C_3_-C_6_. Based on the structure of the flavonoids, they can be classified into six major classes, flavanols (Catechin, Epigallocatechin), flavones (Apigenin and Baicalain), flavonols (Quercetin and Kaempferol), flavanones (Naringenin and Hesperidin), isoflavones (Genistein and Daidzein), and anthocyanins (Cyanidin and Pelargonidin). The best-described property of flavonoids is their antioxidant capacity. These phenolic compounds can delay, inhibit, or prevent oxidation by scavenging free radicals and reducing oxidative stress [60].

The *S. sinensis* extracts contained high yields of flavonoids (4.28 mg/g), and FA (4.13 mg/g) was detected in 75% ethanol combined with ultrasound-assisted extract (UAE). The total antioxidant assay showed superoxide radical-scavenging capacity and in vitro mushroom tyrosinase inhibition in a dose-dependent manner. Based on the results, it was demonstrated that flavonoid compounds and FA from *S. sinensis* that are found in health foods and cosmetic products could have antioxidant properties and inhibit tyrosinase activity [61]. Plant polyphenols and flavonoids are plant metabolites characterized by the presence of several phenol groups that are derived from L-phenylalanine. Flavonoids display a broad range of health-promoting bioactivities [62].

Flavonoids are secondary metabolites that are very abundant in plants. In plants, flavonoids may be found in their free form (aglycones) or linked to sugars. To date, the following flavonoids have been isolated from *S. sinensis*, including 5-hydroxy-3,7,4′-trimethoxyflavone, 3,7-dimethoxy-5-hydroxy-2-{[4-(3-methyl-2-butenyl)oxy]phenyl}-4H-1-benzopyran-4-one, 5-hydroxy-3,7,3′,4′-tetramethoxyflavone, 5-hydroxy-3,7-dimethoxy-4′-(1-hydroxy-3-methylbut-3-en-2-yloxy)-flavone, and 5-hydroxy-4′-[(2-isopentenyl)oxy]- 3,7,3′-trimethoxyflavone [18]. 

Plants are good sources of multifunctional natural molecules. Several natural molecules, including flavonoids, have been reported to exhibit multiple functions. Flavonoids have been shown to have a wide range of anti-cancer properties, including the ability to modulate reactive oxygen species (ROS)-scavenging enzyme activities, participate in cell cycle arrest, induce apoptosis and autophagy, and suppress cancer cell proliferation and invasiveness. The chemical structures of homocyclotirucallane, dihydro/phenanthrenes, 3,4-dihydroxybenzaldehyde (3,4-DHB), FA, and flavonoids, which are the main compounds isolated from *S. sinensis*, are shown in Figure 3 and Table 1.

## 3. Phytochemicals and Natural Plant Compounds

Plant-derived products have the potential to fight against illnesses, with little or no secondary effects compared to synthetic ones. The screening of phytochemical compounds in medicinal and aromatic plants plays a significant role in the human diet, animal feed, pharmaceuticals, fragrances, and cosmetics [63,64]. Plant flavonoid compounds are a unique class of so-called “nutraceuticals” that have the ability to protect against oxidative damage. Phytochemicals are naturally occurring compounds in plants and are a good source of useful hepatoprotective agents that can modulate the activity of free radicals [65,66].

Phytochemicals are secondary metabolites synthesized through various biosynthetic pathways. Currently, plant-based bioactive compounds are widely used as key ingredients in nutraceutical and pharmaceutical products owing to their therapeutic properties. Phytochemicals and other natural products have the potential to promote health and prevent various diseases and have been used in traditional and complementary medicine for many years [67]. Natural plant compounds (NPCs) have been used for many years as sources of therapeutic substances and as structural bases for drug development. NPCs are bioactive elements isolated from natural plants that regulate EMT through anti-inflammatory, anti-fibrotic, and antioxidant mechanisms [68]. Several important drugs used in modern medicine have been reported in medicinal plant studies, including taxol/paclitaxel, vinblastine, vincristine, topotecan, irinotecan, etoposide, and teniposide. Medicinal plants are among the most important sources of drugs worldwide. Plant-derived extracts are naturally preferred by most individuals because they have fewer side effects, are safer and cheaper, and tend to promote health better than most synthetic/semisynthetic drugs. Combining an existing anti-cancer agent with a flavonoid or a flavonoid with other phytochemicals has shown improved efficacy in cancer treatment.

Plants containing a variety of phenolic compounds have been shown to play an important role in cancer prevention because they act as dietary antioxidants. New anti-cancer approaches can be explored using traditional Chinese medicinal (TCM) plants, which are excellent sources of chemotherapeutic agents with various biological activities and great therapeutic value [69,70]. Medicinal plants and natural herbal products are often administered in combination with chemotherapeutic drugs and conventional therapies to increase their potential effects and provide better protection against side effects [71,72,73].

Orchidaceae is one of the largest families of flowering plants with more than 27,000 accepted species, and more than 31,000–35,000 species are estimated to exist in total [74]. Organic compounds from natural sources, such as plants, animals, and microorganisms, have led to drug development [75]. Newman and Cragg concluded that more than 50% of all modern clinical drugs originate from natural products [76]. The use of orchids in herbal medicines has a long history. Orchids have been used as a source of medicine for millennia to treat various diseases [77]. Orchidaceae, being one of the biggest plant families, is recognized as a rich source of bioactive secondary metabolites, which overall, up to now, has been understudied but can offer new opportunities in the search for new classes of bioactive natural products.

## 4. Future Perspectives and Conclusions

Herbal medicine is derived from natural sources and may contain poisonous substances which can affect safety and effectiveness. Natural products and their derivatives have always been the focus of the research and development of new drugs. In this review, we outlined herbal medicines and plant-derived natural products of *Spiranthes sinensis* (Orchidaceae) (Figure 4). The plant extracts and different phytochemicals of *Spiranthes sinensis* are known to have a variety of biological activities at the cell and animal level. Combining an existing anti-cancer agent with a flavonoid or a flavonoid with other phytochemicals has shown improved efficacy in cancer treatment. Orchids, similar to other plants, produce a large number of phytochemicals. Published research on orchid-derived biocompounds has revealed the presence of several phytochemicals. *S. sinensis*, also known as Panlongshen, is a common folk herb with ornamental and medicinal value. This paper provided a narrative review of the phytochemistry and pharmacology of *Spiranthes sinensis*. The information in this work may provide a basis for better exploitation of the medicinal plant value of the genus Orchidaceae in the future.

## Figures and Tables

**Figure 1 plants-11-01692-f001:**
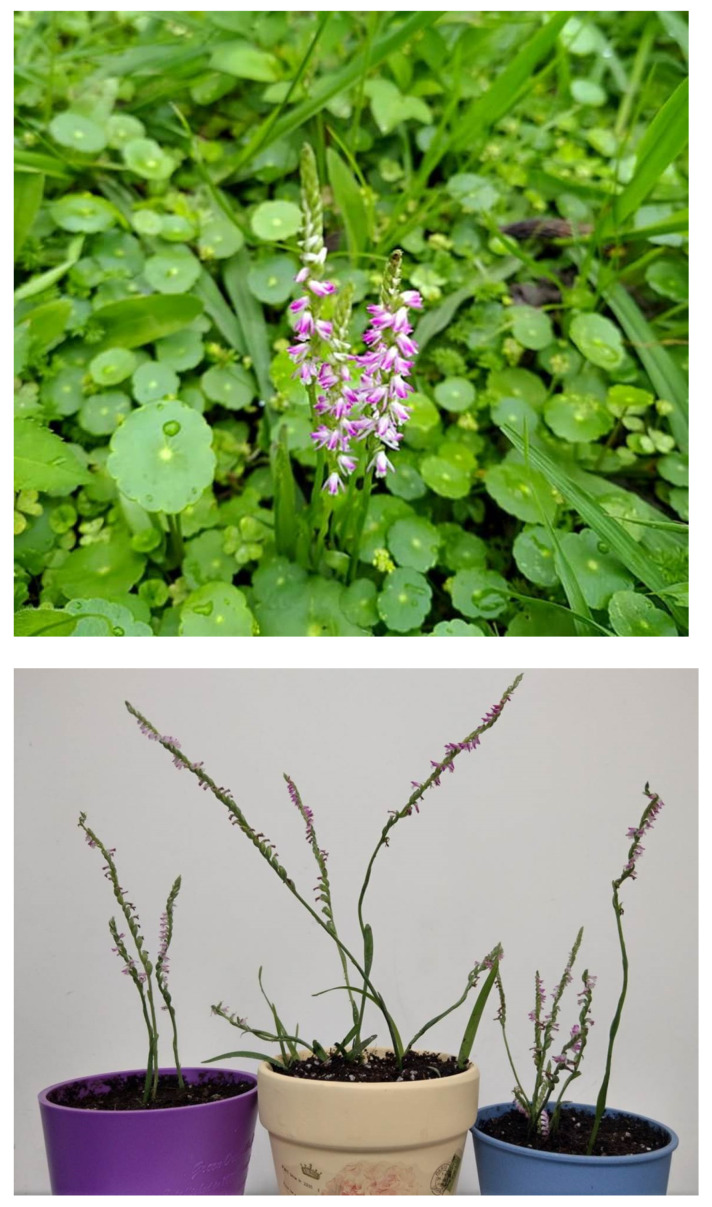
*Spiranthes sinensis* (Shou Tsao) is an orchid plant, its fleshy root is like ginseng, and is also called panlongshen. In the Taiwan climate, it flowers during the Ching Ming Festival, also called the ching ming tsao. Whole *Spiranthes sinensis* (Orchidaceae) plants were collected from Kainan University, Taoyuan City (121°15′00″ E, 25°03′00″ N), Taiwan, by Professor Yu-Jen Kuo. The plant material of *S. sinensis* samples were harvested (April and May 2019).

**Figure 2 plants-11-01692-f002:**
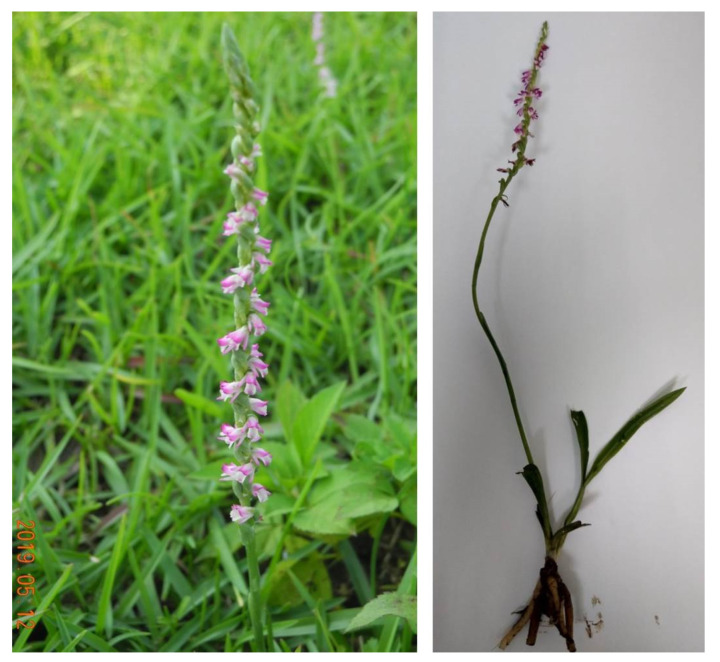
Whole *Spiranthes sinensis* (Orchidaceae) plants were collected from Kainan University, Taoyuan City. *S. sinensis* (pers.) Ames, also known as Panlongshen among the common folk herbs, has ornamental and medicinal value. *S. sinensis* inflorescence and the handedness of the receptacle is shown. The whole plant can be used as a medicine. The wild population of this plant is not easy to find.

**Figure 3 plants-11-01692-f003:**
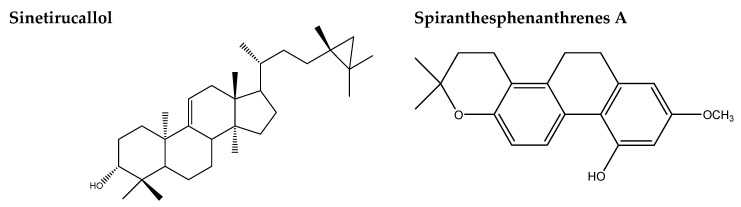
Chemical structures of homocyclotirucallane and dihydro/phenanthrenes, 3,4-dihydroxybenzaldehyde (3,4-DHB), ferulic acid (FA), and flavonoid, which are the main compounds isolated from *S. sinensis*.

**Figure 4 plants-11-01692-f004:**
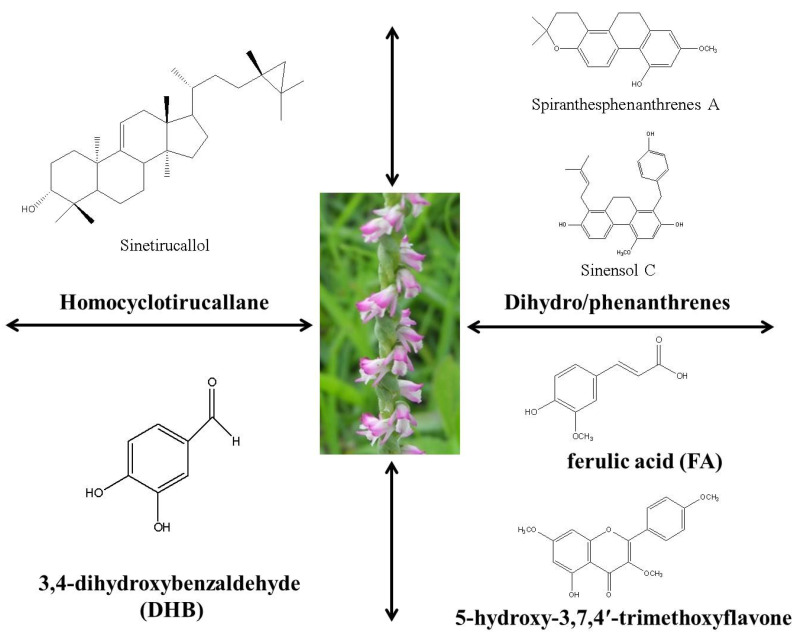
Graphical depiction of *Spiranthes sinensis* and its bioactive compounds in this review article.

**Table 1 plants-11-01692-t001:** Bioactive compounds of *S. sinensis* and their pharmacodynamics.

Bioactive Compound/Extract	Biological Activities	References
**Ethyl acetate fraction**	Anti-inflammatory effects in LPS-stimulated RAW264.7 cells and BALB/c mice	[9]
**Sinetirucallol**	Moderate anti-Hepatitis B virus e antigen (HBeAg) activity in MS-G2 hepatoma cell line	[6,10]
**Sinensol G, H**	Moderate anti-Hepatitis B virus e antigen (HBeAg) activity in MS-G2 hepatoma cell line	[6,10]
**Sinensol C**	Inhibits adipogenesis in 3T3-L1 cells (upregulation of AMPK)	[19]
**Spiranthesphenanthrenes A**	Inhibited B16-F10 cancer cell migration by inhibition of EMT	[18]
**3,4-dihydroxybenzaldehyde**	Anti-neuroinflammatory effects in rat models of middle cerebral artery occlusion/reperfusion and lipopolysaccharide (LPS)-treated BV2 microglial cells	[27]
	Inhibits oxidative DNA damage in human blood cells	[28,29]
**Ferulic acid**	Induces MIA PaCa-2 pancreatic cancer cell apoptosis	[36]
	Induces apoptosis of HeLa and Caski cervical carcinoma cells by downregulating PI3K/Akt pathway	[37]
	Protects PC12 cell and neural stem cell hypoxic injury	[38,39,40]
	Ameliorates cisplatin-induced nephrotoxicity in rats	[41]
	FA inhibits TGF-β1 and drives the EMT process in NRK-52E cells	[42]
	FA and astragaloside IV synergistically reduced renal fibrosis in UUO rats	[43]
	Effective against chronic cerebral hypoperfusion-induced swallowing dysfunction in animals	[45]
	Inhibits lipid peroxidation in STZ-induced diabetes in rats	[46]
	Alleviates CCL4, TGFβ-induced liver fibrosis in mice and hepatic stellate cell activation via inhibition of the TGF-β/Smad signaling pathway	[51,52,53]
	Ameliorates doxorubicin-induced cardiac toxicity in rats	[55]
	Effective against cisplatin-induced ototoxicity in HEI-OC1 cells	[56]
	FA can inhibit the NF-κB/NLRP3 inflammasome axis in MTX-induced nephrotoxicity	[59]
**Flavonoids**	75% ethanol extract of *S. sinensis* obtained the highest yield of flavonoids, and ferulic acid showed high antioxidant activity and inhibited tyrosinase activity	[61]

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
