# Peer review of "Pharmacological and Chemical Potential of Spiranthes sinensis (Orchidaceae): A Narrative Review"

_plants, 2022, doi:10.3390/plants11131692_

Round 1

Reviewer 1 Report

For my oppinion paper entiteld Pharmacological and chemical potential of Spiranthes sinensis (Orchidaceae): A Narrative Review is a comprehensive review with all the essential facts included. Because of that I suggest to accept this paper in present form.

Author Response

Response: We like to thank for reviewer’s opinion and recommendation.

Reviewer 2 Report

In this review the authors tried to describe the chemical and pharmacological application of “Spiranthes sinensis” on the basis of its folklore use and natural products derived from this plant.

1.       First of all, I think the review lacks adequate amounts of references. For making a narrative review the authors should try to include more relevant works as reference to enhance the credibility of the work.

2.       The authors included the chemicals derived from the plant and their application. But in case of all the bioactive chemicals there is no information about the pharmacokinetics and pharmacodynamics of the chemicals.

3.       In the table 1, the authors did not include any information about the types of the study or mechanism of the function. Making this table is not giving any significant information about the pharmacological use of the compopunds.

4.       All the compounds are described to have anticancer potential. However the authors should include a separate segment about the anticancer effect of the compounds with the detailed mechanism. And also on which specific cancer the compound showed activity must be described here.

5.       Also the study lacks the future potential of the compounds in research and medicinal experiments. The author should discuss the potential future studies using this plant to increase the significance of the work.

6.       The authors discussed the impact of the bioactive compounds in ros modulation in this review work. ROS is a very important controller of many cellular signaling pathways. So, the authors should dedicate a separate segment on this mechanism. Also in different places the authors used the abbreviated form “ROS” and in other places the full form. This should be corrected.

Author Response

1. 

Response: The authors like to thank for reviewer’s suggestion. We have modified our manuscript according to the reviewers' comments and suggestions. We have added and modified these points in page 4-6 in the revised manuscript as indicated.

Added refs:

  • Anjum, R.; Maheshwari, N.; Mahmood, R. 3,4-Dihydroxybenzaldehyde mitigates fluoride-induced cytotoxicity and oxidative damage in human RBC. Trace Elem. Med. Biol. 2022, 69, 126888-126896.
  • Maheshwari, N.; Mahmood, R. 3,4-Dihydroxybenzaldehyde attenuates pentachlorophenol-induced cytotoxicity, DNA damage and collapse of mitochondrial membrane potential in isolated human blood cells. Drug Chem. Toxicol. 2022, 45(3), 1225-1242.
  • Wei, M.G.; Sun, W.; He, W.M.; Ni, L.; Yang, Y.Y. Ferulic acid attenuates TGF-ß1-induced renal cellular fibrosis in NRK-52E cells by inhibiting Smad/ILK/Snail pathway. Based Complement. Alternat. Med. 2015, 2015, 619720-619726.
  • Meng, L.Q.; Tang, J.W.; Wang, Y.; Zhao, J.R.; Shang, M.Y.; Zhang, M.; Liu, S.Y.; Qu, L.; Cai, S.Q.; Li, X.M. Astragaloside IV synergizes with ferulic acid to inhibit renal tubulointerstitial fibrosis in rats with obstructive nephropathy. J. Pharmacol. 2011, 162, 1805–1818.

Mahmoud, A.N.; Hussein, O.E.; EI-Twab, S.M.A.; Hozayen, W.G. Ferulic acid protects against methotrexate nephrotoxicity via activation of Nrf2/ARE/HO-1 signaling and PPARg, and suppression of NF-κB/NLRP3 inflammasome axis. Food Funct. 2019, 10, 4593–4607.

2. 

Response: We like to thank for reviewer’s opinion and suggestion. Information on the studies of Spiranthes sinensis was collected from scientific journals, reports via library and electronic data search (PubMed, Elsevier, Google Scholar). The results shown follow: Orchidaceae and Spiranthes sinensis have 15 results. Spiranthes sinensis have 21 results. Spiranthes sinensis and pharmacokinetics had no results were found. Spiranthes sinensis and pharmacodynamics have 4 results. I have already tried my best to added pharmacodynamics of the chemicals in the revised manuscript and Table 1.

3. 

Response: Thanks for your comments, we have agreed the reviewer suggestion. We have already changed and rewrote ‘’Bioactive compounds of S. sinensis and their phaemacodynamics’’ in Table 1 (page 8-9) in the revised manuscript.

4. 

Response: The authors like to thank for reviewer’s reminding. We separate segment about the biological activities of the compounds. We have checked this point and modified this part in the revised manuscript. Such as

2.1.1. Anti-hepatoma cell

2.1.2. Inhibits adipogenesis in 3T3-L1 cells

2.1.3. Anti-cancer activity

2.2.1. Anti-oxidant activity

2.3.1. Anti-inflammatory activity

2.3.2. Apoptosis pathway

2.3.3. Neuroprotective

2.3.4. Anti-nephrotoxicity and anti-diabetes

2.3.5. Anti-hepatic fibrosis

2.3.6. Synergistically effects

2.4.1. Antioxidant activity

5. 

Response: We are grateful to reviewer’s suggestion. We have added and modified these points in page 9-10 in the revised manuscript as indicated.

Future perspectives and conclusions

Herbal medicine is derived from natural sources and may contain poisonous substances which can affect safety and effectiveness. Natural products and their derivatives have always been the focus of research and development of new drugs. In this review, we have outlined herbal medicines and plant-derived natural products of Spiranthes sinensis (Orchidaceae) (Figure 4). The plant extracts and different phytochemicals of Spiranthes sinensis are known to have a variety of biological activities at the cell and animal level. Plant-derived extracts are naturally preferred by most individuals because they have fewer side effects, are safer and cheaper, and tend to promote health better than most synthetic/semisynthetic drugs. Combining an existing anti-cancer agent with a flavonoid or a flavonoid with other phytochemicals has shown improved efficacy in cancer treatment.

Orchids, similar to other plants, produce a large number of phytochemicals. Published research on the orchid-derived biocompounds has revealed the presence of several phytochemicals. S. sinensis also known as Panlongshen, and is a common folk herb with ornamental and medicinal value. This paper provides a narrative review of phytochemistry and pharmacology of the Spiranthes sinensis. The information in this work may provide a basis for better exploitation of the medicinal plant value of the genus Orchidaceae in the future.

6. 

Response: We appreciate the reviewer’s reminding. We have already corrected these points (page 5) in the revised manuscript as indicated.

We sincerely hope that the editor and reviewer are satisfied with our efforts and that these changes and replies meet your requirements for publication of the manuscript.

Reviewer 3 Report

The manuscript “Pharmacological and Chemical Potential of Spiranthes sinensis (Orchidaceae): A Narrative Review” contains interesting information about the composition, bioactivity and herbal medicine formulations and plant-derived natural products of S. sinensis.

The review presents contemporary literature data on wide range of natural compounds and bioactivities in view of the plant of ornamental and medicinal value, widely distributed throughout Asia and Oceania, as well as the rest of the world.

Considering this, I think that the review reported in this paper is relevant and likely to be of great interest to readers of your reputable Journal and therefore I recommend the Editorial board to accept it for publication in present form.

Author Response

Response: We appreciate the reviewer’s opinion and recommendation.

Round 2

Reviewer 2 Report

the authors have improved the manuscript as suggested.